

# Conflicting phylogenetic signals in plastomes of the tribe Laureae (Lauraceae)

Tian-Wen Xiao[1,2], Yong Xu[1,2], Lu Jin[1,2], Tong-Jian Liu[1], Hai-Fei Yan[1] and Xue-Jun Ge[1]

[1] Guangdong Provincial Key Laboratory of Applied Botany and Key Laboratory of Plant Resources Conservation and Sustainable Utilization, South China Botanical Garden, Chinese Academy of Sciences, Guangzhou, Guangdong, People's Republic of China
[2] University of Chinese Academy of Sciences, Beijing, People's Republic of China

## ABSTRACT

**Background**. Gene tree discordance is common in phylogenetic analyses. Many phylogenetic studies have excluded non-coding regions of the plastome without evaluating their impact on tree topology. In general, plastid loci have often been treated as a single unit, and tree discordance among these loci has seldom been examined. Using samples of Laureae (Lauraceae) plastomes, we explored plastome variation among the tribe, examined the influence of non-coding regions on tree topology, and quantified intra-plastome conflict.

**Results**. We found that the plastomes of Laureae have low inter-specific variation and are highly similar in structure, size, and gene content. Laureae was divided into three groups, subclades I, II and III. The inclusion of non-coding regions changed the phylogenetic relationship among the three subclades. Topologies based on coding and non-coding regions were largely congruent except for the relationship among subclades I, II and III. By measuring the distribution of phylogenetic signal across loci that supported different topologies, we found that nine loci (two coding regions, two introns and five intergenic spacers) played a critical role at the contentious node.

**Conclusions**. Our results suggest that subclade III and subclade II are successively sister to subclade I. Conflicting phylogenetic signals exist between coding and non-coding regions of Laureae plastomes. Our study highlights the importance of evaluating the influence of non-coding regions on tree topology and emphasizes the necessity of examining discordance among different plastid loci in phylogenetic studies.

## INTRODUCTION

Gene tree discordance is relatively common in phylogenomic studies. The conflicts can be caused by biological factors like incomplete lineage sorting (ILS), hybridization, horizontal gene transfer, gene loss, and gene duplication (*Maddison, 1997*; *Sun et al., 2015*; *Gonçalves et al., 2019*; *Sato et al., 2019*). Most relevant studies have focused on incongruent tree topologies among different genomic compartments (*Sun et al., 2015*; *Zhao et al., 2016*; *Walker et al., 2019*) because these genes have evolved independently and their gene

Corresponding author
Xue-Jun Ge, xjge@scbg.ac.cn

tree topologies have been influenced by biological processes. By contrast, relatively few studies have focused on tree conflicts among plastid genes (e.g., *Foster, Henwood & Ho, 2018*; *Gonçalves et al., 2019*; *Walker et al., 2019*; *Zhang et al., 2020*). Usually, plastomes are considered to be uniparentally inherited and to have evolved as a single unit, free from such biological sources of conflict (*Birky, 1995*; *Wicke et al., 2011*). However, the branched and linear structure of plastid DNA, which arose from recombination-dependent replication, is indicative of recombination (*Oldenburg & Bendich, 2016*; *Ruhlman et al., 2017*). In addition, biparental inheritance and heteroplasmy (e.g., the presence of different plastomes within an individual or a cell) have been reported in seed plants (*Szmidt, Aldén & Hällgren, 1987*; *Johnson & Palmer, 1989*; *Reboud & Zeyl, 1994*; *Carbonell-Caballero et al., 2015*). Heteroplasmy may, in rare cases, give rise to heteroplasmic recombination, which has been invoked to explain gene tree discordance (*Marshall, Newton & Ritland, 2001*; *Sullivan et al., 2017*; *Sancho et al., 2018*). In addition to recombination events, the transfer of genes among plastid, mitochondrial and nuclear genomes; positive selection; tree length (a proxy for evolutionary rate); and GC content may also generate phylogenomic conflict (e.g., *Stegemann et al., 2003*; *Smith, 2014*; *Wysocki et al., 2015*; *Piot et al., 2018*; *Saarela et al., 2018*; *Foster, Henwood & Ho, 2018*). Aside from biological factors, non-biological factors (e.g., outlier genes, uninformative loci, and gaps) may cause conflict as well. For example, *Duvall, Burke & Clark (2020)* found that alternative topologies arose from alignment gaps. Given that most studies assume no conflict and treat the plastome as a single unit, taking biological and non-biological factors into consideration and quantifying the extent of conflict among different plastid loci is of great importance (*Wolfe & Randle, 2004*).

Owing to the rapid development of next-generation sequencing (NGS), more plastomes are becoming available at a reasonable cost, driving advances in phylogenomics and promoting a more comprehensive understanding of plant evolution (*Li et al., 2019*). Phylogenetic relationships among Lauraceae (*Song et al., 2017*), as well as many other groups (e.g., *Eserman et al., 2014*; *Barrett et al., 2016*), have been well resolved using plastome data. In phylogenomic studies of plastomes (*Guo et al., 2017*; *Gonçalves et al., 2019*; *Xu et al., 2019*; *Li et al., 2019*), plastome coding genes have generally been used, and non-coding regions have been excluded. Only a few studies have noted the potential impact of non-coding regions on tree topology. *Parks, Cronn & Liston (2009)* revealed that the phylogenetic position of *Pinus albicaulis* Engelm. based on complete plastomes differed from that based on exon sequences. A similar situation also occurred for phylogenetic relationships within Rubiaceae (*Wikström, Bremer & Rydin, 2020*), suggesting that there were conflicting phylogenetic signals between coding- and non-coding regions. Because tree topology is the foundation of comparative studies that infer biogeographic history, phylogenetic diversity and other evolutionary patterns (*Walker et al., 2019*), the influence of non-coding regions on phylogenetic inference should be evaluated.

Both ILS and hybridization are at play in tree species, which generally have high rates of outcrossing and large population sizes (*Petit & Hampe, 2006*; *Crowl et al., 2019*). Interspecific hybrids have been described in *Persea* (tribe Perseeae, sister to tribe Cinnamomeae and tribe Laureae), *Cinnamomum* and *Aiouea* (tribe Cinnamomeae) (*van der Werff, 1984*; *Rohwer et al., 2019*). These processes are perhaps also problematic

in Laureae. When combined, such biological processes may make accurate inference of evolutionary relationships in Laureae difficult. Unfortunately, previous phylogenomic studies of Laureae have ignored potential conflicts among different plastid loci and the underlying processes that may have generated them (*Zhao et al., 2018*; *Song et al., 2019*; *Tian, Ye & Song, 2019*). These characteristics make Laureae an ideal group in which to explore intra-plastome conflict and its influence on phylogenetic inference.

Tribe Laureae, a species-rich group in the family Lauraceae, is phylogenetically sister to tribe Cinnamomeae (*Song et al., 2019*). It comprises approximately 500 species and 10 genera: *Actinodaphne*, *Adenodaphne*, *Dodecadenia*, *Iteadaphne*, *Laurus*, *Lindera*, *Litsea*, *Neolitsea*, *Parasassafras* and *Sinosassafras* (*Van der Werff & Richter, 1996*; *Chanderbali, Van der Werff & Renner, 2001*; *Li et al., 2004*; *Li et al., 2008b*). Species of this tribe are evergreen or deciduous and usually occur in the form of trees or shrubs (*Li et al., 2008a*). Their distribution ranges from the Mediterranean region, Asia, and Oceania to North America (*Li et al., 2004*). Some members of Laureae have great ecological and economic value. For example, *Neolitsea sericea* (Bl.) Koidz. is a dominant species found in various evergreen and deciduous broadleaf mixed forests and in evergreen broadleaf forests (*Wang et al., 2009*), and *Laurus nobilis* L. has been used in remedies for centuries (*Nayak et al., 2006*).

Although Laureae is monophyletic, generic delimitation within this tribe remains unclear (*Kostermans, 1957*; *Hutchinson, 1964*; *Li et al., 2008b*). *Adenodaphne*, endemic to New Caledonia, is closely related to *Litsea* (*Chanderbali, Van der Werff & Renner, 2001*). However, morphological confusion still exists between this genus and *Litsea*, meaning that their distinctiveness and the monophyly of *Adenodaphne* require further study (*Chanderbali, Van der Werff & Renner, 2001*). *Actinodaphne* is polyphyletic and closely related to the monophyletic genus *Neolitsea* (*Li et al., 2007*; *Li et al., 2008b*; *Fijridiyanto & Murakami, 2009a*, *Fijridiyanto & Murakami, 2009b*). Although *Fijridiyanto & Murakami (2009a)* and *Fijridiyanto & Murakami (2009b)* argued that *Actinodaphne* was monophyletic, the species of *Actinodaphne* sampled in their analyses were totally different from those sampled in *Li et al. (2007)* and *Li et al. (2008b)*. Furthermore, *Lindera* and *Litsea* have been shown to be polyphyletic, with *Dodecadenia*, *Iteadaphne*, *Laurus*, *Parasassafras* and *Sinosassafras* nested within them (*Li et al., 2004*; *Li et al., 2008b*). *Liu et al. (2017)* used three plastid barcode loci combined with the internal transcribed spacer (ITS) region for species identification and found that the Laureae tree was polytomic. Despite these efforts, phylogenetic relationships among and within these genera have been poorly resolved based on molecular markers like the ITS, the external transcribed spacer (ETS), *matK*, *trnL-F* and *trnH*-*psbA*. Compared with these molecular markers, complete plastomes have better performance at the species level within Laureae, although generic delimitation still remains unclear due to limited taxon sampling (*Zhao et al., 2018*; *Song et al., 2019*; *Tian, Ye & Song, 2019*).

Thirty-five plastomes representing 28 species and six genera of Laureae have been published (Table S1). Compared with the vast diversity of Laureae, the published plastome data for this group are relatively limited. Hence, we now report 12 newly sequenced plastomes (Table 1) and combine them with existing plastomes to address three primary

**Table 1  Sampled species and voucher specimens of Laureae in this study.**

| Taxon | Herbarium | Voucher | Geographic origin | GenBank Accession number |
|---|---|---|---|---|
| *Actinodaphne obovata* (Nees) Bl. | IBSC | XTBGLQM0236 | Xishuangbanna, Yunnan, China | MN274947 |
| *Iteadaphne caudata* (Nees) H. W. Li | IBSC | XTBGLQM0582 | Xishuangbanna, Yunnan, China | MN428456 |
| *Lindera erythrocarpa* Makino | IBSC | 180923 | Baishanzu Mountain, Zhejiang, China | MN428457 |
| *Litsea acutivena* Hay. | \ | \ | Chebaling, Guangdong, China | MN428458 |
| *Litsea dilleniifolia* P. Y. Pai et P. H. Huang | IBSC | XTBGLQM0095 | Xishuangbanna, Yunnan, China | MN428459 |
| *Litsea elongata* (Wall. ex Nees) Benth. et Hook. f. | IBSC | WBGQXJ001 | Badagong Mountain, Hunan, China | MN428460 |
| *Litsea glutinosa* (Lour.) C. B. Rob. | IBSC | XTBGLQM0653 | Xishuangbanna, Yunnan, China | MN428461 |
| *Litsea mollis* Hemsl. | IBSC | CFL2678 | Libo county, Guizhou, China | MN428462 |
| *Litsea monopetala* (Roxb.) Pers. | IBSC | XTBGLQM0687 | Xishuangbanna, Yunnan, China | MN428463 |
| *Litsea pungens* Hemsl. | IBSC | WBGQXJ124 | Badagong Mountain, Hunan, China | MN428464 |
| *Litsea szemaois* (H. Liu) J. Li et H.W. Li | IBSC | XTBGLQM0692 | Xishuangbanna, Yunnan, China | MN428465 |
| *Neolitsea pallens* (D. Don) Momiy. et H. Hara | IBSC | 18371 | Dinghu Mountain, Guangdong, China | MN428466 |

goals: (1) reinvestigation of phylogenetic relationships within Laureae; (2) examination of conflict between coding and non-coding regions; and (3) quantification of conflicts among different plastid loci.

## METHODS

### Plant materials, DNA extraction and genome sequencing

Materials from 12 species in five genera (*Actinodaphne obovata* (Nees) Bl., *Iteadaphne caudata* (Nees) H. W. Li, *Lindera erythrocarpa* Makino, *Litsea acutivena* Hay., *L. elongata* (Wall. ex Nees) Benth. et Hook. f., *L. glutinosa* (Lour.) C. B. Rob., *L. dilleniifolia* P. Y. Pai et P. H. Huang, *L. mollis* Hemsl., *L. monopetala* (Roxb.) Pers., *L. pungens* Hemsl., *L. szemaois* (H. Liu) J. Li et H.W. Li, and *Neolitsea pallens* (D. Don) Momiy. et H. Hara) (tribe Laureae, Lauraceae) were collected and identified by the authors (Table 1). Voucher specimens were deposited in the herbarium of the South China Botanical Garden (IBSC) at the Chinese Academy of Sciences. No specific permissions were required for the relevant locations and activities. Including the plastomes downloaded from GenBank and the Lauraceae Chloroplast Genome Database (LCGDB, https://lcgdb.wordpress.com) (Table S1), this study included 47 Laureae plastomes, representing seven genera and all subclades identified by *Song et al. (2019)*. Twelve plastomes from other tribes were also downloaded (Table S1).

Genomic DNA was extracted from silica-gel-dried leaf tissue using the cetyl trimethyl ammonium bromide (CTAB) method (*Doyle & Doyle, 1987*). The yields of genomic DNA extracts were quantified by fluorometric quantification on a Qubit instrument (Invitrogen, Carlsbad, California, USA) using the dsDNA HS kit, and the DNA size distribution was assessed visually on a 1% agarose gel. DNA libraries with an average insert size of 270 bp were prepared by the Beijing Genomics Institute (BGI, Shenzhen, China). Paired-end reads of $2\times 151$ bp were generated on the Illumina X ten sequencing system (Illumina Inc.).

## Plastid genome assembly, annotation and comparison

Low-quality reads and adaptors were removed using Trimmomatric v0.36 (*Bolger, Lohse & Usadel, 2014*), generating approximately 3 Gb of high-quality clean reads per sample. The clean reads were analyzed for quality control with FastQC (*Andrews, 2010*) and then used to assemble plastomes with NOVOPlasty v2.7.2 (*Dierckxsens, Mardulyn & Smits, 2016*). To guarantee assembly quality, clean reads were mapped to the assembled plastid genomes using the Burrows-Wheeler Aligner (BWA 0.7.17-r1188 (*Li & Durbin, 2010*)) and samtools 1.9 (*Li et al., 2009*), and were visually checked in Geneious Prime 2019.1.

Plastome annotation was performed using the program GeSeq - Annotation of Organellar Genomes (*Tillich et al., 2017*). Start and stop codons were inspected and manually adjusted in Geneious Prime when necessary. Plastomes were submitted to GenBank (MN274947, MN428456–MN428466). Maps of all 12 plastomes were drawn using the OrganellarGenomeDRAW tool (OGDRAW) (*Lohse et al., 2013*). A summary of the newly sequenced plastomes is presented in Table 2.

To illustrate interspecific sequence variation within Laureae, plastomes of *A. obovata*, *I. caudata*, *Laurus nobilis* (KY085912), *Lindera erythrocarpa*, *Litsea acutivena*, *N. pallens* and *Parasassafras confertiflorum* (Meisn.) D. G. Long (MH729378) were aligned using MAFFT (*Katoh & Standley, 2013*) with default settings. Sequence identity was plotted with the mVISTA program using the LAGAN mode (*Frazer et al., 2004*), with *Lindera glauca* (Siebold et Zucc.) Bl. (MF188124) as a reference.

## Phylogenetic reconstruction and tests for selection

To evaluate potential conflicts, phylogenetic trees were constructed using maximum likelihood (ML) methods based on six datasets: (1) complete plastome (CP), (2) coding regions (CDS), (3) non-coding regions (non-CDS), (4) large single copy region (LSC), (5) small single copy region (SSC), and (6) one inverted repeat region (IR).

Sequences were aligned using MAFFT with default settings and manually edited with BioEdit v7.2.5 (*Hall, 1999*) when necessary. The best-fitting DNA substitution models for the six unpartitioned datasets were selected using ModelTest-NG (*Darriba et al., 2020*) under the corrected Akaike Information Critierion (AICc). The aligned sequences and selected DNA substitution models were used for ML analyses, and ML trees were constructed using RAxML-NG (*Kozlov et al., 2019*). We also implemented a partitioning strategy on two datasets, the CP with one IR region removed (CP-reduced) and CDS (configuration details shown in File S1). The optimal partitioning schemes for each dataset were inferred with PartitionFinder 2 (*Lanfear et al., 2016*), and the optimal partitioning schemes, and nucleotide substitution models for each partition were used for ML analyses in RAxML-NG.

Because gaps can affect tree topology (*Duvall, Burke & Clark, 2020*), we also performed the following analysis based on the CP dataset. 'Mask Alignment' in Geneious Prime was used to strip the gaps from the MAFFT alignment, with the threshold set to 0 (no gaps), 2%, 10%, 20%, 50% or 75%. The resulting alignments were used to infer ML trees in RAxML-NG.

**Table 2  Summary of 12 complete plastomes of Laureae.**

| | Actinodaphne obovata | Iteadaphne caudata | Lindera erythrocarpa | Litsea acutivena | Litsea elongata | Litsea glutinosa |
|---|---|---|---|---|---|---|
| Total cpDNA size (bp) | 152,579 | 152,863 | 152,916 | 152,718 | 152,793 | 152,748 |
| Length of LSC region (bp) | 93,655 | 93,761 | 93,921 | 93,677 | 93,827 | 93,698 |
| Length of IR region (bp) | 20,057 | 20,144 | 20,071 | 20,066 | 20,066 | 20,062 |
| Length of SSC region (bp) | 18,810 | 18,814 | 18,853 | 18,909 | 18,844 | 18,926 |
| Total GC content (%) | 39.1 | 39.1 | 39.1 | 39.2 | 39.1 | 39.2 |
| LSC GC content (%) | 37.9 | 38.0 | 37.9 | 38.0 | 37.9 | 38.0 |
| IR GC content (%) | 44.4 | 44.4 | 44.4 | 44.4 | 44.4 | 44.5 |
| SSC GC content (%) | 33.9 | 33.8 | 34.0 | 33.9 | 33.9 | 33.8 |
| Total number of genes (unique) | 127 (112) | 127 (112) | 127 (112) | 127 (112) | 127 (112) | 127 (112) |
| Protein-coding genes (unique) | 84 (78) | 84 (78) | 84 (78) | 84 (78) | 84 (78) | 84 (78) |
| Total number of tRNA | 36 (30) | 36 (30) | 36 (30) | 36 (30) | 36 (30) | 36 (30) |
| Total number of rRNA | 8 (4) | 8 (4) | 8 (4) | 8 (4) | 8 (4) | 8 (4) |

| | Litsea dilleniifolia | Litsea mollis | Litsea monopetala | Litsea pungens | Litsea szemaois | Neolitsea pallens |
|---|---|---|---|---|---|---|
| Total cpDNA size (bp) | 152,298 | 152,736 | 152,705 | 152,655 | 152,132 | 152,699 |
| Length of LSC region (bp) | 93,218 | 93,655 | 93,758 | 93,520 | 93,119 | 93,761 |
| Length of IR region (bp) | 20,094 | 20,063 | 20,074 | 20,131 | 20,090 | 20,071 |
| Length of SSC region (bp) | 18,892 | 18,936 | 18,799 | 18,873 | 18,843 | 18,796 |
| Total GC content (%) | 39.2 | 39.2 | 39.2 | 39.2 | 39.2 | 39.1 |
| LSC GC content (%) | 38.0 | 38.0 | 38.0 | 37.9 | 38.1 | 37.9 |
| IR GC content (%) | 44.4 | 44.4 | 44.4 | 44.4 | 44.4 | 44.4 |
| SSC GC content (%) | 34.0 | 33.9 | 33.9 | 34.0 | 34.0 | 33.9 |
| Total number of genes (unique) | 127 (112) | 127 (112) | 127 (112) | 127 (112) | 127 (112) | 127 (112) |
| Protein-coding genes (unique) | 84 (78) | 84 (78) | 84 (78) | 84 (78) | 84 (78) | 84 (78) |
| Total number of tRNA | 36 (30) | 36 (30) | 36 (30) | 36 (30) | 36 (30) | 36 (30) |
| Total number of rRNA | 8 (4) | 8 (4) | 8 (4) | 8 (4) | 8 (4) | 8 (4) |

Positive selection on plastid coding genes has the potential to bias phylogenies (e.g., *Piot et al., 2018*; *Saarela et al., 2018*), and we therefore performed positive selection tests using CODEML in PAML 4.9j (*Yang, 2007*). Coding genes were extracted and aligned in Geneious Prime using MAFFT, stop codons were removed manually, and the aligned sequences were converted to PAML format. Because site models allow dN/dS ratio to vary among different sites, we implemented M0, M1a, M2a, M3, M7 and M8 to identify positively selected sites. Likelihood ratio tests (LRTs) were performed using pchisq function in R 3.6.2 (*R Core Team, 2018*) to test if there was significant difference between models (M0 vs M3, M2a vs M1a, M8 vs M7). We manually deleted positively selected sites when LTRs was significant (M2a vs M1a and/or M8 vs M7 with $p$ value less than 0.05). Coding gene alignments with positively selected sites removed were concatenated (CDS-reduced dataset), and used for ML tree inference to examine whether positive selection can bias phylogeny or not.

## Node support investigation and tree topology tests

Because gene contents were not identical among Cryptocaryeae, *Cassytha*, *Caryodaphnopsis*, *Neocinnamomum* and other clades, the following analyses were performed using a dataset from which six plastomes had been removed (*Beilschmiedia pauciflora* H. W. Li, *Caryodaphnopsis malipoensis* Bing Liu et Y. Yang, *Cassytha filiformis* L., *Cryptocarya chinensis* (Hance) Hemsl. and *Eusideroxylon zwageri* Teijsm. et Binn.).

We extracted all loci (coding regions, introns, tRNA, rRNA and intergenic spacers) using a python script (*Jin, 2019*) and aligned them using MAFFT with default settings. These alignments were used to infer gene trees by rapid bootstrap analyses (option -f a) in RAxML (*Stamatakis, 2014*) with the GTRGAMMA model. The number of bootstrap replicates was set to 1000, as *Simmons & Kessenich (2019)* have suggested that fewer replicates may be insufficient to find the optimal gene tree topology. The best-scoring ML trees were used to estimate the species tree with local posterior probability (LPP) (*Sayyari & Mirarab, 2016*) in ASTRAL III (*Zhang et al., 2018*).

We performed constrained maximum likelihood analyses in IQ-TREE (*Nguyen et al., 2014*) to obtain the ML trees that supported different topologies. To understand which loci supported the alternative topologies, we calculated site-wise log-likelihood values for each topology in RAxML using option "-f G". After obtaining site-wise lnL differences, we converted site-wise differences to locus-wise lnL differences ($\Delta$lnL) in R 3.6.2. The lnL differences were plotted against each locus using ggplot2 (*Wickham, 2016*). It has been suggested that loci with an absolute $\Delta$lnL > 2 are statistically significant (*Edwards, 1984*). Therefore, we conducted separate ML analyses on datasets from which these loci (absolute $\Delta$lnL >2) had been removed to test whether small subsets of sequence matrices determined tree topology (*Shen, Hittinger & Rokas, 2017*).

The Kishino–Hasegawa test (KH test) (*Kishino & Hasegawa, 1989*), Shimodaira-Hasegawa test (SH test) (*Shimodaira & Hasegawa, 1999*) and Approximately-Unbiased test (AU test) (*Shimodaira, 2002*) were used in IQ-TREE to assess the statistical significance of incongruence based on complete plastomes (including only one copy of the IR regions). We specified 10,000 RELL (resampling of estimated log-likelihoods) replicates for the topological tests.

## RESULTS

### Plastome features of Laureae

The sizes of the 12 newly generated Laureae plastid genomes ranged from 152,132 bp (*Litsea szemaois*) to 152,916 bp (*Lindera erythrocarpa*) (Table 2), similar to previously published Laureae plastomes (152,211–153,011 bp, Table S1). All had a typical quadripartite structure and were assembled into a single, circular and double-stranded DNA sequence (Fig. 1). The length of the LSC, SSC and IR regions ranged from 93,119 bp (*Litsea szemaois*) to 93,921 bp (*Lindera erythrocarpa*), 18,796 bp (*N. pallens*) to 18,936 bp (*Litsea mollis*), and 20,057 bp (*A. obovata*) to 20,144 bp (*I. caudata*), respectively, with little variation in size (Table 2). The overall GC contents ranged from 39.1% to 39.2%. GC content was unequally distributed within the plastomes; it was highest in IR regions (44.4–44.5%), moderate in LSC regions (37.9–38.1%), and lowest in SSC regions (33.8–34.0%, Table 2).

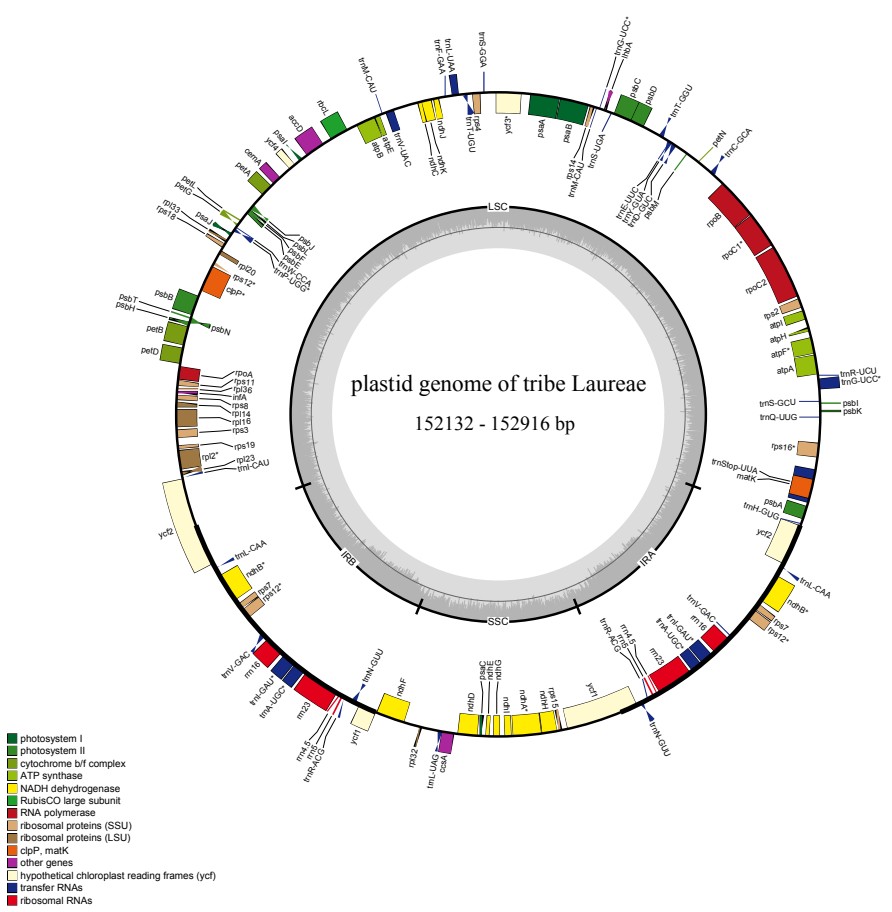

**Figure 1 Complete plastid genome map of Laureae.** Different functional genes are color coded. Genes outside the circle are transcribed counterclockwise, genes inside the circle are transcribed clockwise. GC content is indicated by darker gray in the inner circle.

The 12 newly sequenced plastomes contained 112 single-copy genes: 78 protein-coding genes, 30 tRNA genes, and 4 rRNA genes (Table 2 and Table S2). Sixteen genes had one intron, and two genes had two introns. There were 13 duplicated genes in the IR regions (Table S2), and *rps12*, *ycf1*, and *ycf2* were partly duplicated in the IR regions (Fig. 1).

## Phylogenetic reconstruction and positive selection tests

The GTR+I+G4 model was selected for the six unpartitioned datasets (CP, CDS, non-CDS, LSC, SSC and IR). Perseeae was sister to Cinnamomeae and Laureae (Fig. 2 and Figs. S1–S5). All the ML trees indicated the monophyly of Laureae with high bootstrap (BS) support values (99–100%, Fig. 2 and Figs. S1–S4), except for the ML tree based on the IR region (71%, Fig. S5). This result was caused by the low variability of the IR region (Fig. S6). In the five ML trees (Fig. 2 and Figs. S1–S4), Laureae was divided into three groups. Subclade I included *Lindera communis* Hemsl., *L. glauca* and *L. nacusua* (D. Don) Merr.; subclade II included *Laurus azorica* (Seub.) Franco, *L. nobilis*, *Lindera megaphylla* Hemsl., *Litsea acutivena*, *L. glutinosa*, *L. monopetala* and *L. pungens*; and subclade III
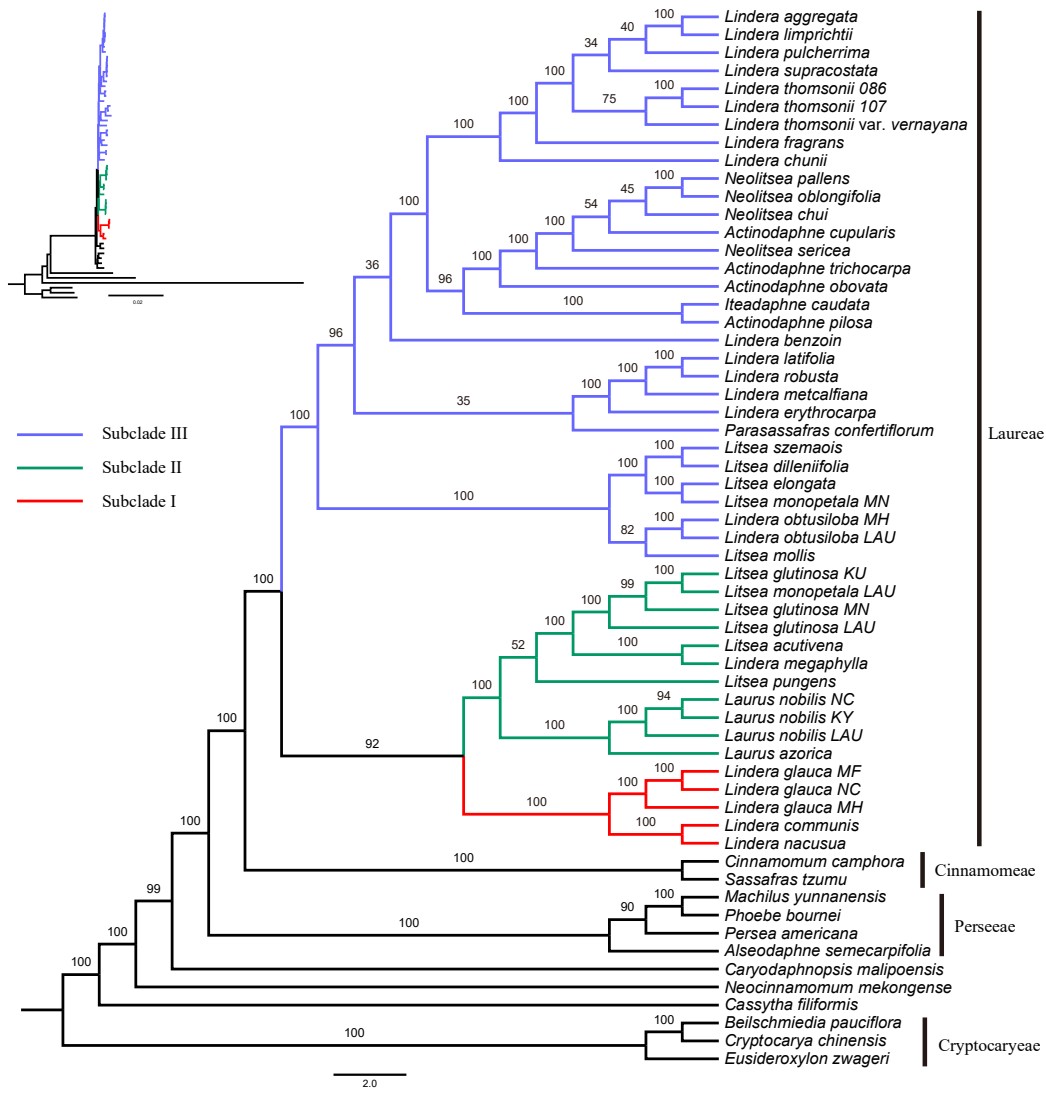

**Figure 2** **Maximum likelihood phylogenetic tree of Laureae inferred with RAxML-NG based on complete plastomes.** Bootstrap values are indicated on branches. Subclades I, II and III are colored in red, green and blue, respectively.

included the other Laureae species used in the study. In subclade I, *Lindera glauca* was sister to *L. communis* and *L. nacusua*. In subclade II, *Laurus* was sister to *Litsea acutivena*, *L. glutinosa* and *Lindera megaphylla*, and the position of *Litsea pungens* was unstable (Fig. 2 and Figs. S1–S4). *Litsea monopetala* (LAU00063) was embedded within three samples of *Litsea glutinosa* in subclade II, highlighting the necessity of re-identification for *L. monopetala* (LAU00063). Topologies within subclade III based on different datasets were largely congruent (Fig. 2 and Figs. S1–S4). In subclade III, samples of *Litsea*, together with *Lindera obtusiloba* Bl., were monophyletic. *Lindera erythrocarpa*, *L. latifolia* Hook. f., *L. metcalfiana* Allen and *L. robusta* (Allen) Tsui were monophyletic as well. *Lindera aggregata*,

*L. chunii* Merr., *L. fragrans* Oliv., *L. limprichtii* H. Winkl., *L. pulcherrima* (Wall.) Benth., *L. supracostata* Lec., *L. thomsonii* Allen and *L. thomsonii* var. *vernayana* (Allen) H.P. Tsui formed a well-supported clade. *Neolitsea* was closer to *Actinodaphne* than to other Laureae species.

Subclade II was sister to subclade I based on four unpartitioned datasets (CP, non-CDS, LSC, SSC; Fig. 2 and Figs. S2–S4, respectively). However, subclade II was sister to subclade III rather than subclade I based on the unpartitioned CDS dataset (Fig. S1). Both topologies were strongly supported. The sister relationship of subclades I and II was supported in the ML tree based on partitioned plastomes (one IR removed, CP-reduced dataset; Fig. S7), and subclade II was sister to subclade III in the ML tree based on the partitioned CDS dataset (Fig. S8), indicating that our results were robust to different partitioning schemes.

The sister relationship of subclades I and II (BS values ranging from 80% to 92%) was consistently revealed even as the percentage of gaps increased (Table S3), indicating that gaps had no impact on our tree topology. Positively selected sites were detected in 27 coding genes (Table S4). The ML tree based on CDS-reduced dataset supported a sister relationship of subclades II and III (Fig. S9), consistent with ML trees based on CDS dataset (Figs. S1 and S8), suggesting that positive selection did not affect the relationship of the three subclades.

## Investigating incongruent nodes and differences in tree topology

The tree topology inferred from ASTRAL III (Fig. 3) was largely congruent with that of the ML trees (Fig. 2 and Figs. S1–S4), except that the former showed a sister relationship of subclade I and subclade III. We performed constrained maximum likelihood analyses in IQ-TREE and obtained three suboptimal ML trees that supported the subclade II–subclade I (called T1 hereafter), subclade II–subclade III (T2) and subclade I–subclade III (T3) affinities. We extracted 243 loci and assessed how each locus supported one of the three topologies by examining the gene-wise log-likelihoods (Fig. 4). T1 was strongly supported by six loci (*rpoC1* intron, *trnG-trnfM*, *ndhA* intron, *psaJ-rpl33*, *rpl2-rpl23* and *petN-psbM*; absolute ΔlnL >2); T2 was strongly supported by three loci (*psaB*, *trnS-ycf3* and *ycf2*; absolute ΔlnL >2); and T3 was moderately supported by one locus (*clpP* intron1; absolute ΔlnL >1 and <2) (Table S5). The sum of absolute ΔlnL of T1 was higher than that of T2 and T3 (Fig. 4), suggesting that our data support the topology of T1 rather than T2 or T3. After the removal of six loci (*rpoC1* intron, *trnG-trnfM*, *ndhA* intron, *psaJ-rpl33*, *rpl2-rpl23* and *petN-psbM*), a sister relationship of subclade II and subclade III was revealed (Fig. S10). After the removal of three loci (*psaB*, *trnS-ycf3*, and *ycf2*), subclade II was sister to subclade I (Fig. S11). These results underscore the decisive role played by small subsets of loci in phylogenetic inference.

The topological tests showed that T2 did not differ significantly from T1 ($p > 0.05$, Table S6). T3 was statistically rejected by the KH and AU tests ($p < 0.05$) but not by the Shimodaira-Hasegawa (SH) test ($p = 0.0505$). That T3 was rejected according to the KH and AU tests suggests that the sister relationship between subclades I and III may be misleading.

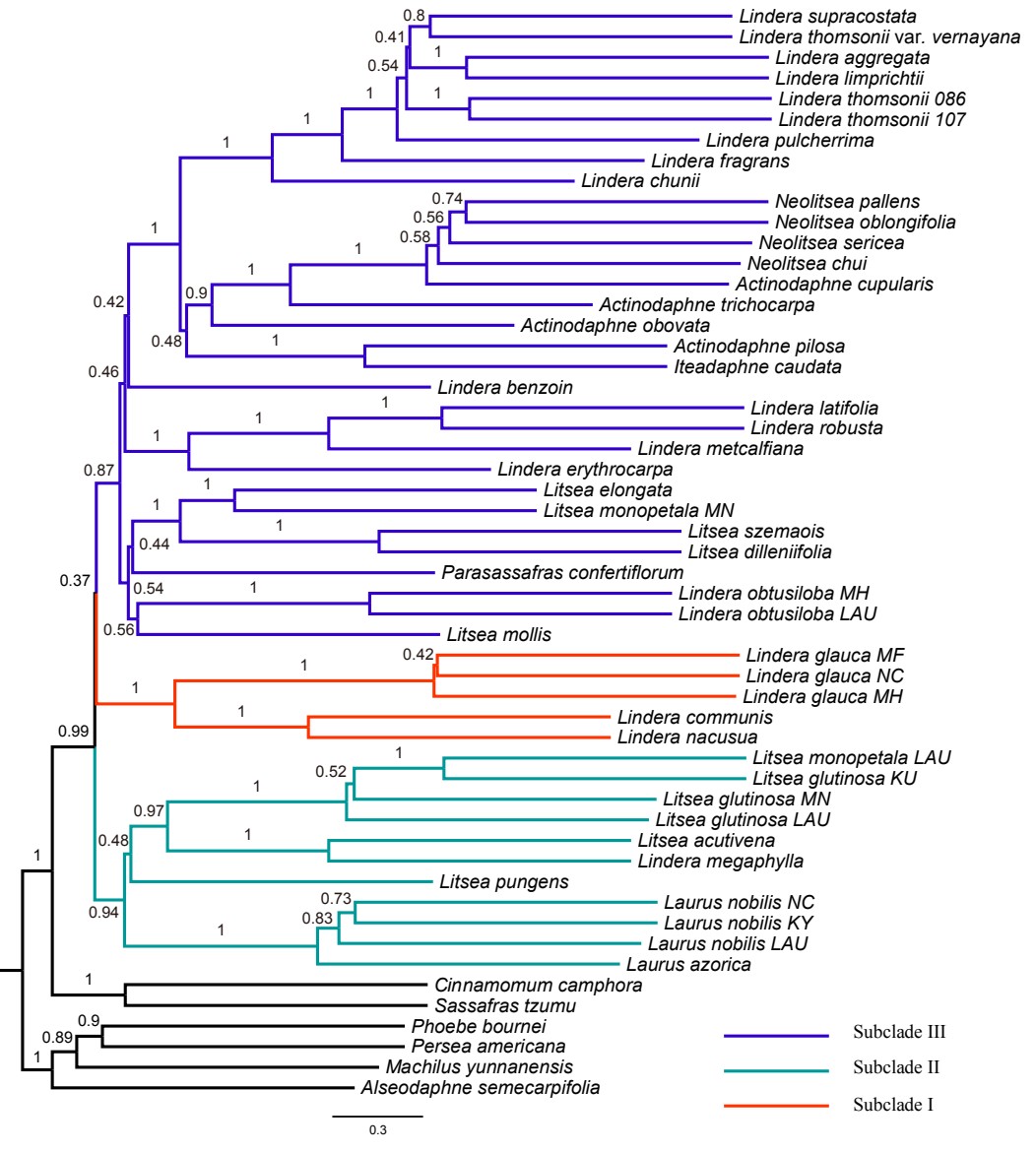

**Figure 3** **Tree of Laureae inferred with ASTRAL III using a multispecies coalescent approach.** Local posterior probabilities (LPP) are indicated on branches. Subclades I, II and III are colored in red, green and blue, respectively.

## DISCUSSION

### Plastome features

It has been noted that most plastid genomes of land plants and algae range from 120 to 160 kilobase pairs (kb) in length (*Palmer, 1985*). In this study, the plastid genome sizes of 12 species from five Laureae genera ranged from 152,132 bp to 152,916 bp, indicating that plastid genome size was conserved within Laureae. GC content was highest in the IR region rather than in the single copy regions, owing to the presence of a ribosomal RNA gene

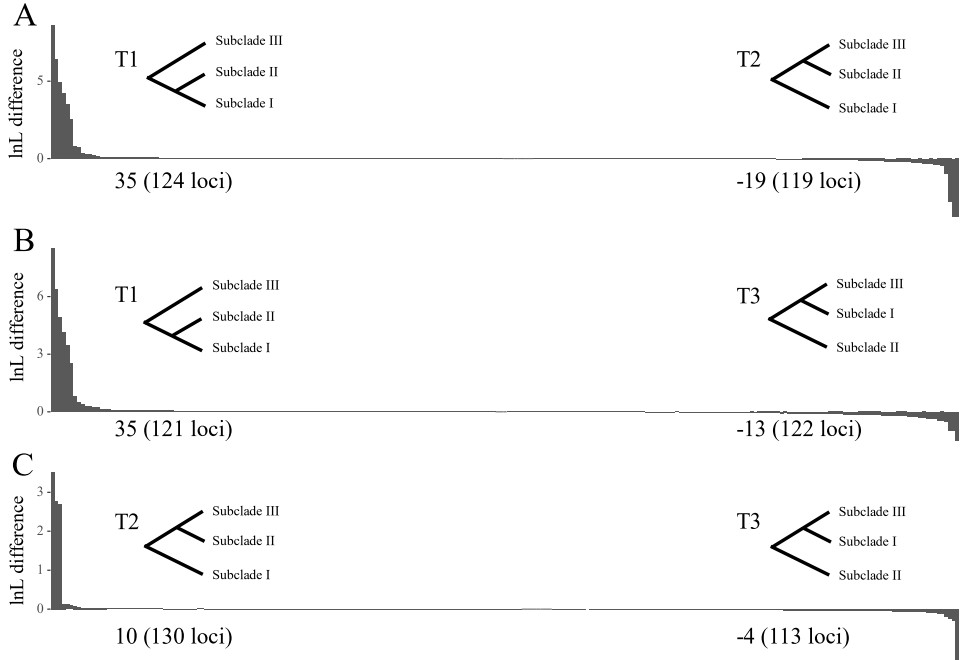

**Figure 4 Difference in the log-likelihood (lnL) of each plastid locus between two alternative topologies.** The *x* axis indicates each locus, and the *y* axis indicates lnL difference. (A) Positive and negative values support the topology showing subclades I–II (T1) and subclades II–III (T2), respectively. (B) Positive and negative values support the topology showing subclades I–II (T1) and subclades I–III (T3), respectively. (C) Positive and negative values support the topology showing subclades II–III (T2) and subclades I–III (T3), respectively. Values starting with + or – indicate the sum of positive and negative values, respectively, and the number of supporting loci is shown in the parenthesis. Note that the order of loci on *x* axis are different among A, B and C.

cluster in the IR region, consistent with a previous study (*Huotari & Korpelainen, 2012*). GC contents of the IR, LSC and SSC regions of the newly sequenced plastomes were identical to those of nine *Lindera* species studied earlier (*Zhao et al., 2018*). In contrast to the gene losses recognized in several Lauraceae lineages (*Song et al., 2017*), our analysis revealed that gene content among Laureae was highly conserved. *Song et al. (2017)* suggested that plastome contraction in Lauraceae was largely driven by fragment loss events in the IR regions. In our study, we found no gene loss among Laureae plastomes.

## Phylogenetic relationships within Laureae

Previous phylogenetic studies (*Song et al., 2017*; *Zhao et al., 2018*) based on complete plastomes suggested that Laureae was sister to Cinnamomeae and that together they were sister to Perseeae. The same phylogenetic relationships among these groups were recognized in our study (Figs. 2 and 3). In previous work, *Actinodaphne* and *Neolitsea* were resolved as monophyletic groups based on *matK*, ITS and rpb2 (*Fijridiyanto & Murakami, 2009a*; *Fijridiyanto & Murakami, 2009b*), but *Actinodaphne* was not a monophyletic group based on complete plastid genomes (*Song et al., 2019*). In this study, the non-monophyletic status of *Actinodaphne* was supported. The conclusion of *Actinodaphne* monophyly may

have been caused by sampling bias in previous studies (*Fijridiyanto & Murakami, 2009b*; *Fijridiyanto & Murakami, 2009a*). The monophyly of *Neolitsea* can be neither rejected nor supported in the present study. *Actinodaphne cupularis* (Hemsl.) Gamble was grouped with *Neolitsea oblongifolia* Merr. et Chun, *N. pallens* and *N. chui* Merr. with low bootstrap support (54%; Fig. 2), and sampling of *Neolitsea* and related genera was limited. *Lindera* and *Litsea* were polyphyletic in our analysis, consistent with previous studies (*Li et al., 2008b*; *Fijridiyanto & Murakami, 2009b*). The phylogenetic position of *P. confertiflorum* was unresolved based on ETS and ITS (*Li et al., 2008b*), and the ambiguity of its position still remains, despite the integration of complete plastid genomes in our analysis and a previous study (*Liao, Ye & Song, 2018*).

Subclade III was sister to subclade I and II in our study, consistent with previous analyses (*Zhao et al., 2018*; *Song et al., 2019*; *Tian, Ye & Song, 2019*). The three *Lindera* species in subclade I share common morphological traits, such as alternate and pinninerved leaves, a persistent involucre, vegetative terminal buds in inflorescences and 3-merous flowers (*Li et al., 2008a*). However, these characters also occur in several members of the other two subclades (e.g., *Lindera benzoin* (L.) Bl. and *Laurus nobilis*), perhaps resulting from convergent and/or parallel evolution (*Li et al., 2008b*). These traits are not good indicators for delimiting the three subclades of Laureae. In subclade III, the trinerved or triplinerved species of *Lindera* (*Lindera aggregata*, *L. chunii*, *L. fragrans*, *L. limprichtii*, *L. pulcherrima*, *L. supracostata*, *L. thomsonii* and *L. thomsonii* var. *vernayana*) formed a well-supported clade in both our study and that of *Tian, Ye & Song (2019)*. However, triplinerved leaves also exist in most species of *Neolitsea* (*Li et al., 2008b*; *Li et al., 2008a*). Therefore, traditional morphological traits are of limited use in taxon delimitation, even within subclades of Laureae. Given the limited samples and data in our analyses, more sampling and DNA sequences are needed to further elucidate the relationships within Laureae.

## Phylogenetic incongruence in the plastome

Although many studies have treated plastid protein-coding genes or the complete plastome as a single unit (e.g., *Song et al., 2019*; *Tian, Ye & Song, 2019*), potential conflicts among sequence types (i.e., coding vs. non-coding regions) have been reported in several studies. By comparing phylogenies based on complete plastomes and coding regions (*Yu et al., 2017*), it was inferred that non-coding regions did not significantly influence the tree topology of Theaceae. By contrast, non-coding regions had an impact on the phylogenetic relationships of several tribes in Rubiaceae (*Wikström, Bremer & Rydin, 2020*) and subtribes in Poaceae (*Saarela et al., 2018*). A conflicting signal between coding and non-coding regions was also reported in Leguminosae (*Zhang et al., 2020*). In this study, inclusion of non-coding regions altered tree topology in the tribe Laureae, suggesting the existence of a conflicting signal between coding and non-coding regions. Non-coding regions are often discarded for being uninformative, or for being misleading due to saturation at deep time scales (*Foster, Henwood & Ho, 2018*). In our study, tree topologies based on coding and non-coding regions were largely congruent, except for the relationships among the three subclades (Figs. S1–S2), indicating that non-coding regions are as informative as coding regions in Laureae. Thus, it is imperative to evaluate the influence of non-coding regions on tree

topology rather than treating the whole plastome as a single unit or simply excluding non-coding regions from phylogenetic analysis.

To accommodate the conflicts among different plastid regions, a species tree was inferred through summary coalescent analysis. It has been suggested that the coalescent method is more robust than the concatenation method when the level of ILS is high (*Liu, Xi & Davis, 2014*; *Mirarab, Bayzid & Warnow, 2014*). High ILS tends to occur when the time interval between consecutive speciation events is short (*Sun et al., 2015*; *Sato et al., 2019*), and the core Lauraceae group (Perseeae, Cinnamomeae and Laureae) is thought to have undergone a rapid radiation (*Chanderbali, van der Werff & Renner, 2001*; *Rohwer & Rudolph, 2005*; *Nie, Wen & Sun, 2007*). We therefore chose to implement the coalescent method. Nonetheless, it should be noted that, with this method, short and uninformative loci may lead to problematic gene trees and therefore result in a less accurate species tree (*Xi, Liu & Davis, 2015*; *Springer & Gatesy, 2016*). In our study, only nine of 243 loci (*rpoC1* intron, *trnG-trnfM*, *ndhA* intron, *psaJ-rpl33*, *rpl2-rpl23*, *petN-psbM*, *psaB*, *trnS-ycf3*, and *ycf2*) had a strong phylogenetic signal at the contentious node. The other 234 loci with weak phylogenetic signals may have resulted in gene trees with uncertainties and led to inaccurate topology at this node.

Exploration of the factors that underlie conflicts in phylogenetic signals is of great importance—but it is also challenging. Previous studies have examined whether biological and non-biological factors contribute to such conflicts (e.g., *Duvall, Burke & Clark, 2020*; *Zhang et al., 2020*). For example, gaps have been found to cause alternate, but conflicting topologies in Poaceae (*Duvall, Burke & Clark, 2020*). However, the inclusion of alignment gaps did not alter our tree topology (Table S3). Although previous studies indicated that partitioning improves phylogenetic inference (*Xi et al., 2012*), ML tree topologies based on partitioned and unpartitioned datasets did not differ significantly in our study. It has been suggested that plastid genes under positive selection may bias phylogenies (e.g., *Piot et al., 2018*; *Saarela et al., 2018*), however, we found that the relationship among subclades I, II and III was not affected by positively selected sites, suggesting that positive selection was not the cause of tree conflicts. In this study, the low support values and short branch lengths of the estimated species tree (Fig. 3) suggested that each locus had a significantly incongruent topology and may indicate the existence of ILS. High levels of ILS are thought to yield similar numbers of loci supporting alternative topologies (*Huson et al., 2005*). In our study, the numbers of loci supporting each topology were different (six for T1, three for T2, and zero for T3 after exclusion of loci with absolute $\Delta lnL \leq 2$), suggesting that ILS may not be the primary cause of the discordance among loci. Another plausible explanation for the conflict is heteroplasmic recombination, which can occur in species with biparental plastome inheritance (*Walker et al., 2019*). Although heteroplasmic recombination has been reported with clear evidence in *Brachypodium* and *Picea* (*Sullivan et al., 2017*; *Sancho et al., 2018*), to our knowledge it has never been documented in Lauraceae. Based on the data reported here, it is too early to draw a firm conclusion about the causes of the conflict in phylogenetic signals. Although fully resolved phylogenies may still remain elusive based on different genomic compartments (i.e., nuclear, mitochondrial and plastid), phylogenomic studies that incorporate these compartments can provide new insights into tree discordance

and its underlying causes (*Koenen et al., 2020*). Therefore, more genetic information (e.g., nuclear genes) will be required to solve this problem in future work.

## CONCLUSION

In summary, this study revealed that Laureae plastomes are conserved in structure, size and gene content. A conflicting phylogenetic signal was detected between coding and non-coding regions, suggesting that the plastid genome should not be treated as a single unit. ML trees based on coding and non-coding regions were largely congruent except at the contentious node, indicating that coding regions are as informative as non-coding regions and that the influence of non-coding regions on tree inference should be evaluated. We also found that small subsets of plastome loci determined the topology at specific nodes, consistent with the results of a previous study (*Shen, Hittinger & Rokas, 2017*). Through quantification and analysis of intra-plastome conflicts, the sister relationship of subclade I (including *Lindera communis*, *L. glauca* and *L. nacusua*) and II (including *Laurus azorica*, *L. nobilis*, *Lindera megaphylla*, *Litsea acutivena*, *L. glutinosa*, *L. monopetala* and *L. pungens*) was supported by our study. Biological factors may contribute to the conflicts among plastid loci; however, more information is needed to determine the underlying mechanism(s).

## ACKNOWLEDGEMENTS

The authors thank Yu-Ying Zhou and Chen-Xin Ma for assistance with the DNA experiments. They also thank Feng Song for her kind help in plastid genome assembly and annotation. The authors would like to thank TopEdit for English language editing of this manuscript.

### Funding

This study was financially supported by the Strategic Priority Research Program of the Chinese Academy of Sciences, Grant No. XDB31000000. There was no additional external funding received for this study. The funders had no role in study design, data collection and analysis, decision to publish, or preparation of the manuscript.

### Grant Disclosures

The following grant information was disclosed by the authors:
Strategic Priority Research Program of the Chinese Academy of Sciences: XDB31000000.

### Competing Interests

The authors declare there are no competing interests.

### Author Contributions

- Tian-Wen Xiao conceived and designed the experiments, analyzed the data, prepared figures and/or tables, authored or reviewed drafts of the paper, and approved the final draft.

- Yong Xu and Lu Jin analyzed the data, prepared figures and/or tables, and approved the final draft.
- Tong-Jian Liu and Hai-Fei Yan analyzed the data, authored or reviewed drafts of the paper, and approved the final draft.
- Xue-Jun Ge conceived and designed the experiments, authored or reviewed drafts of the paper, and approved the final draft.

## DNA Deposition

The following information was supplied regarding the deposition of DNA sequences:

The 12 newly sequenced plastomes are available at GenBank: MN274947, MN428456–MN428466.

## Data Availability

The 12 newly sequenced plastomes in our study are available in Data S1.

Voucher specimens were deposited in the herbarium of the South China Botanical Garden (IBSC) and can be searched by collection number: XTBGLQM0236, XTBGLQM0582, 180923, XTBGLQM0095, WBGQXJ001, XTBGLQM0653, CFL2678, XTBGLQM0687, WBGQXJ124, XTBGLQM0692, 18371(see also Table 1).

## Supplemental Information

Supplemental information for this article can be found online at http://dx.doi.org/10.7717/peerj.10155#supplemental-information.

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
