# Peer review of "Conflicting phylogenetic signals in plastomes of the tribe Laureae (Lauraceae)"

_PeerJ, doi:10.7717/peerj.10155_

## Round 0.1 · original submission · Minor Revisions

Thank you for making the improvements to your manuscript. there remain a small number of issues to address, particularly the possible impact of selection bias on tree reconstruction. There are some minor corrections still to make on the grammar in the manuscript. If you can address these issues I do not expect to send this our for any further review before acceptance.

Reviewer 1 ·

Basic reporting

No comment.

Experimental design

No comment.

Validity of the findings

No comment.

Additional comments

Following are my comments on manuscript #49532 entitled “Conflicting phylogenetic signals in plastomes of the tribe Laureae (Lauraceae)” by Xiao and coworkers.

Note that special formatting (e.g., italics) may not be preserved in this online review.

Line 42 The authors fail to include one major biological factor that has the potential to introduce artifacts to phylogenies. This is natural selection, especially positive selection, of protein coding sequences. Selection is well-documented for plastid genes and has the potential to bias phylogenies (e.g., see Piot et al., 2018; Saarela et al. 2018).

Lines 52-53 The phrase: “…biparental inheritance and heteroplasmy…has been reported…” should be “…biparental inheritance and heteroplasmy…have been reported…”

Lines 72-73, 330-332 A recent paper, in which the impact of noncoding partitions in the plastomes of Poaceae were investigated, is Saarela et al. (2018). The authors should cite this relevant paper and compare their methods and results with those of Saarela et al. (2018).

Lines 99-100 The phrase: “…Neolitsea sericea is a constructive species …” is unclear and should be reworded. Perhaps the intended meaning was that N. sericea is a source of lumber used for construction.

Line 124 The phrase: “…we now report 12 new plastomes…” should be “…we now report 12 newly sequenced plastomes…”

Lines 232-233 This sentence regarding model selection may be more appropriate in the “Methods” section of the manuscript.

Line 362 The phrase: “…cause alternate but conflicting topologies…” should be “…cause alternate, but conflicting topologies…” (insert comma).

Lines 273-275 When small subsets of genes have a “decisive role” in the inference of phylogenies (line 274), the possibility exists that selection in one or two genes (in this case, ycf2 and psaB) may bias the result creating false phylogenetic signal. The authors should acknowledge this possibility.

The original release date for the paper by Duvall et al. was 2019. However, the paper wasn’t officially published in Botanical Journal of the Linnean Society until 2020. The date of the citations (lines 62, 180, 361, 363) and the reference (line 456) should probably be changed to 2020.

I am unable to find legends for supplemental figures S1-S8.

This paper is clearly written, well-organized, generally well-researched, and uses contemporary methods to examine the plastome phylogenomics of Laureae. The influence of different data partitions on the phylogeny is explored including subsets of coding and noncoding regions and gapped positions introduced by the alignment.

One factor that is overlooked in this otherwise thorough study is the possibility of selection bias. Selection, and the potential impact of positive selection artifacts on phylogenetic results, would ideally be measured for the protein coding sequences, but at least should be discussed.

Literature cited:

Piot, A., Hackel, J., Christin, P.A. and Besnard, G., 2018. One-third of the plastid genes evolved under selection in PACMAD grasses. Planta, 247(1): 255-266.

Saarela, Jeffery M., et al. A 250 plastome phylogeny of the grass family (Poaceae): topological support under different data partitions. PeerJ 6 (2018): e4299.

Reviewer 2 ·

Basic reporting

The manuscript was a pleasure to read, with clear, easily understood English throughout. The study was well placed in the broader context of the field, and, as a result, the purpose of the study is well defined. The raw data are available from GenBank, and the results of each analysis in the study are available either in the main text or the supplement. I have only minor suggestions for additional reporting.

The figure captions could be more complete by describing the phylogenetic trees in full. The ML trees should mention the program that they were inferred with. The ASTRAL tree should mention that ASTRAL the tree was inferred using a multispecies coalescent approach in ASTRAL. Given that many trees were inferred in the study, the reason for presenting those two particular trees in the main text should be justified.

The authors point out that: “By contrast, relatively few studies have focused on tree conflicts among plastid genes”, and “Aside from biological factors, non-biological factors (e.g., outlier genes, uninformative loci, and gaps) may cause conflict as well.” These are good points, and the authors have cited several of the few recent studies that address conflict among plastid genes. However, an additional, relevant citation would be: Foster et al. (2018) Plastome sequences and exploration of tree-space help to resolve the phylogeny of riceflowers (Thymelaeaceae: Pimelea), Molecular Phylogenetics and Evolution, 127: 156-167. In Foster et al. (2018), a topology-clustering approach was used, finding discordance among chloroplast gene trees. The discordance was then investigated with respect to strength of selection (dN/dS), tree length, and GC content. These are useful biological reasons for discordance among chloroplast genes that should be mentioned in the present study.

I found the following sentence a little confusing: “The sum of absolute ΔlnL of T1 was higher than that of T2 and T3 (Fig. 4), suggesting that our data support the topology of T1 rather than T2 or T3.” If it could be rephrased to better explain the rationale behind justifying T1 as better based on likelihood ratios, that would be beneficial.

Finally, it should be mentioned in the discussion that, at deep timescales, non-coding genes are often discarded for being uninformative, or for being misleading due to saturation.

Experimental design

The research is original and within the scope of PeerJ. The objectives of the study are well defined, and represent clear advances on previous work. The Methods are generally described well, with the exception of a few queries.

The authors describe six main, initial data sets for phylogenetic analysis: “(1) complete plastome (cp), (2) coding regions (CDS), (3) non-coding regions (non-CDS), (4) large single copy region (LSC), (5) small single copy region (SSC), and (6) inverted repeat region (IR)”. As the name of the IR implies, it contains two copies of its constituent genes. I’m assuming that only one copy of each gene was used for phylogenetic analysis, but this should be made explicit in the manuscript.

All sequences for each data set were aligned, then analysed using RAxML-NG to estimate trees. A key part of multilocus phylogenetic analysis is partitioning of the data, but there is no mention of partitioning in the manuscript. For example, the CDS data set could/should have been partitioned by gene and by codon position, with appropriate nucleotide substitution models applied to each partition.

Was any partitioning carried out? If yes, please report it in the manuscript. If not, why not? Given the impact of adequate partitioning on topology inference, it’s an important omission.

Validity of the findings

By and large, the findings are valid and well justified, and the conclusions are appropriate given the results of the study. The only minor concern I have about the validity of the findings relates to my query in the experimental design feedback section about partitioning. Depending on how the authors respond to my queries, these concerns may not be important.

Additional comments

Congratulations on a great study.

---

## Round 0.2 · Minor Revisions

While I have indicated minor revisions here, these revisions are important and non-trivial. Reviewer 2 has made some crucial further suggestions that i would urge you to take notice of.

Reviewer 1 ·

Basic reporting

No comment.

Experimental design

No comment.

Validity of the findings

No comment.

Additional comments

In this revision of manuscript #49532 by Xiao and coworkers, the suggestions on the previous draft of the manuscript have been satisfactorily addressed.

No further revisions are needed.

Reviewer 2 ·

Basic reporting

In my previous review, I noted that the manuscript was well written. This feedback still stands, but I have these additional minor comments based on new text:

59: change “gene evolutionary rate” to “a proxy for evolutionary rate:

186: personal preference, but change “cp” to “CP” here (and henceforth). This would also require changing “cp-reduced” to “CP-reduced”. I feel capitalising the data set name helps to delimit the abbreviation in the text, rather than having it look like a typo.

194–195: change to “the CP dataset”

196: Change to “the optimal partitioning schemes for each dataset”.

197: change “the best partition schemes and models for each partition“ to “the optimal partitioning schemes, and nucleotide substitution models for each partition,”

212: change “paml” to “PAML”

293: change “confirmed” to “supported”

295–296: change “indicating that partitioning did not affect our tree topology.” to “indicating that our results were robust to different partitioning schemes”.

293–302: instead of having three one-sentence paragraphs, merge all into one paragraph. The following sentence also needs to be reworded for clarity: “LRT showed that the dN/dS ratios of labeled lineages (subclades I, II and III) were not significantly different from background (p > 0.05), of which dN/dS ratios were less than one (Table S4), suggesting that there was no positive selection on the plastid genes.” I suggest something like:

“Additionally, a likelihood ratio test showed that the dN/dS ratios of labeled lineages (subclades I, II and III) were not significantly different from background (p > 0.05), of which dN/dS ratios were less than one (Table S4), suggesting that there was no positive selection on the plastid genes.”

Please also note my reservations that I have about the selection analyses (including the way they are reported), outlined in the next section of the review.

Experimental design

I’m pleased the authors conducted further analyses using different partitioning schemes and investigating signatures of selection in the protein-coding genes. However, I am not convinced that the selection analyses were conducted properly.

212–215: The explanation of the tests for selection is very confusing: “To statistically test for positive selection, we compared the performance of two branch models (M0 and M2) for each gene. Three foreground branches were labeled on the unpartitioned CDS ML tree. Likelihood ratio tests (LRT) were performed using pchisq function in R 3.6.2 (R Core Team, 2018).”

In a study like this one, it would be useful to test whether there is evidence for selection in any of the protein-coding genes. Such an approach might show that (e.g.) most protein-coding genes are evolving under purifying selection, but a handful are evolving under positive selection. Most plastome studies take this approach. However, the authors in the present study chose to test foreground vs background lineages using branch models in CODEML. This approach is useful to determine whether particular lineage(s) are evolving under different regimes of selection, and can be interesting when there is an a priori explanation of why particular lineages might be expected to be evolving differently from others. To include the selection analyses in the manuscript, the authors need to either (a) justify why CODEML branch models were used, and what particular lineages were selected as foreground lineages and why (in the methods, not just results); or (b) use site models in CODEML to test for global signals of selection in genes based on multiple sequence alignments, rather than comparing particular lineages to a background.

I think the discrepancy in finding no genes under positive selection in the present study, compared to previous studies, might be because of the choice of tests conducted in CODEML.

Regardless of whether the authors take this advice into account, they really need to explain their analysis more carefully so it makes sense. Generally, the authors should also use “positive selection” instead of “natural selection”. It might also be worth mentioning the possibility of “relaxed selection”, since most protein-coding genes should be under purifying selection.

Validity of the findings

I am happy with the validity of the phylogenetic findings. However, I am not convinced of the validity of the selection tests findings (see previous section of review).

---

## Round 0.3 · accepted · Accept

Thank you for making the last revisions requested and dealing with the positively selected sites in an appropriate manner in the analysis. I am happy that your manuscript is now suitable to be accepted.